# Isolated Forest-Based Prediction of Container Resource Load Extremes

**Chaoxue Wang and Zhenbang Wang ***

School of Information and Control Engineering, Xi'an University of Architecture and Technology, Xi'an 710055, China
* Correspondence: ae@wangzhenbang.com; Tel.: +86-181-9172-9187

**Abstract:** Given the wide application of container technology, the accurate prediction of container CPU usage has become a core aspect of optimizing resource allocation and improving system performance. The high volatility of container CPU utilization, especially the uncertainty of extreme values of CPU utilization, is challenging to accurately predict, which affects the accuracy of the overall prediction model. To address this problem, a container CPU utilization prediction model, called ExtremoNet, which integrates the isolated forest algorithm, and classification sub-models are proposed. To ensure that the prediction model adequately takes into account critical information on the CPU utilization's extreme values, the isolated forest algorithm is introduced to compute these anomalous extreme values and integrate them as features into the training data. In order to improve the recognition accuracy of normal and extreme CPU utilization values, a classification sub-model is used. The experimental results show that, on the AliCloud dataset, the model has an $R^2$ of 96.51% and an MSE of 7.79. Compared with the single prediction models TCN, LSTM, and GRU, as well as the existing combination models CNN-BiGRU-Attention and CNN-LSTM, the model achieves average reductions in the MSE and MAE of about 38.26% and 23.12%, proving the effectiveness of the model at predicting container CPU utilization, and provides a more accurate basis for resource allocation decisions.

**Keywords:** cloud computing; CPU prediction; cloud platform resource allocation; cloud platform CPU load prediction; temporal convolutional network





## 1. Introduction

With the continuous development of Internet technology, cloud computing has become a core computing model in modern information technology [1]. In this context, containerization technologies such as Docker have been widely used on cloud platforms [2], providing a more flexible and efficient mode of service deployment for both enterprises and individuals. In cloud computing environments, accurately predicting the extreme values of container CPU utilization is crucial to safeguarding system performance and maintaining efficient resource allocation.

Extreme fluctuations in container CPU utilization may not only lead to unoptimized resource allocation, which reduces the quality of service, but can also trigger system stability issues and severe service disruptions [3,4]. In addition, container CPU utilization prediction provides a key reference for load balancing and container scaling strategies [5], enabling tasks to be dynamically adjusted across multiple containers or computing nodes, thus ensuring the stable and efficient operation of the entire cloud platform. Therefore, accurately predicting container extreme values in CPU utilization has become an important challenge [4].

In order to satisfy users' needs for the high availability, performance, and efficiency of computing services, load forecasting is employed to predict future container CPU usage trends in a given period by analyzing historical usage data [6]. Several methods have been

used for container CPU utilization prediction, including autoregressive moving average models (ARIMAs), Bayesian models, and recurrent neural networks (RNNs); however, they still struggle to deal with the high volatility and rare extreme values in container data, such as container CPU utilization. The existing models often struggle to accurately capture and predict these critical extreme fluctuations, which affects the accuracy of the prediction results and the generalization ability of the model. In addition, maintaining computational efficiency and adapting to variable load characteristics are also challenges hindering existing methods [7]. Therefore, there is an urgent need to further optimize the existing models or explore new strategies to enhance the prediction accuracy.

In this paper, we propose a CPU utilization prediction model, named ExtremoNet, which incorporates isolated forest and classification sub-models to address the extreme value problem in container CPU utilization prediction. The model accurately identifies CPU utilization extreme values by introducing the isolated forest algorithm, and incorporates their abnormal scores as important features in the prediction framework, effectively enhancing the model's responsiveness to extreme situations. Combined with the classification sub-model, it further improves the accuracy of distinguishing between normal CPU utilization and extreme CPU utilization.

This paper is organized as follows: Section 2 introduces the current state of the research on CPU load prediction; Section 3 introduces the ExtremoNet model structure; Section 4 describes the experimental process and related analyses; Section 5 summarizes the paper and proposes the next research direction.

## 2. Related Work

In recent years, many researchers have carried out extensive and in-depth explorations of the container load prediction problem, using a range of methods ranging from traditional statistical methods to modern machine learning and deep learning techniques.

(1) Traditional methods

Y. Xie et al. [8] proposed a hybrid linear model based on ARIMA and triple exponential smoothing. This method performs well when working with small volumes of data or data of high quality, but cannot sufficiently capture all the fluctuating characteristics present in time series data, nor can it effectively deal with the extreme values of the nonlinearities. Joshi, N.S. et al. [9] proposed a method that combines predictive analytics with a theoretical controller, a novel approach that introduces a PID controller to ensure the stability of the system. However, this study assumes that the workload is predictable and stable, which may not hold true in a dynamic environment.

(2) Machine learning methods

Gopal et al. [10] proposed a Bayesian network model based on user behavior and time frame analysis, which takes into account the interactions between concurrent applications while making predictions. The results show that the method has better accuracy compared to the regression and support vector techniques. However, the authors ignored the impact of VM interference on application performance and data center load. R. Hu et al. [11] proposed a CPU load prediction method based on support vector regression (SVR) and Kalman smoothing. Although SVR is excellent at dealing with nonlinear problems, this model performs poorly when the data are unstable or contain a large number of random variations. Zhong et al. [12] proposed a model that combines the advantages of weighted wavelet support vector machine (WWSVM) with PSO. The importance of multiple sample points is considered by weighting the data to reduce the prediction error.

(3) Deep learning methods

Patel et al. [13] proposed a multi-step CPU utilization prediction method called RCP-CL to model continuous and discontinuous CPU load values with random fluctuations and novel continuous and periodic patterns by combining the 1D-CNN (1D-CNN) and LSTM networks. However, this method is only applicable to CPU loads with random fluctuations

and novel continuous and periodic patterns, and the prediction accuracy may be poor in scenarios featuring sudden changes in CPU utilization. Javad Dogani et al. [14] used wavelet transform to decompose the input data into sub-bands with different frequencies to extract patterns from nonlinear and nonsmooth data, thus improving the prediction accuracy, and then input the extracted features into BiGRU to predict future workloads. Devi et al. [15] used a hybrid ARIMA-ANN model to predict future CPU and memory utilization, but this algorithm is limited by the flaws inherent to the ARIMA and ANN models themselves, which hinder their ability to fit nonlinear data. Yifei Wang et al. [16] used Gram's angle field (GAF) by converting the time series data into image format data and extracted the spatio-temporal features using CNN and LSTM; however, the conversion process leads to a loss of information, which affects the accuracy of the prediction results. Malik et al. [17] investigated the prediction of multi-resource utilization using functional link neural networks (FLNNs) combined with genetic algorithms (GAs) and particle swarm optimization (PSO). Genetic algorithms and particle swarm optimization have a slow search process, which may lead to a low computational efficiency. X. Li et al. [18] proposed a combined model based on bi-directional long short-term memory (BILSTM) networks and gated recurrent units (GRUs). However, due to the high impact of outliers on the MSE, this may lead to an incomplete or inaccurate assessment of the model's performance. Lu Wang et al. [19] proposed a load prediction model, CNN-BiGRU-Attention, and a container scheduling strategy based on load prediction. Wang Enxu et al. [20] proposed a dual-attention mechanism network. This network introduces a feature attention mechanism and a temporal attention mechanism on top of LSTM to enhance the importance of feature and temporal information. Cao Zhen et al. [21] proposed an n-LSTM model based on LSTM layers and input features that are extended horizontally. A specialized LSTM model is designed for each feature to be learned. This method relies too much on feature selection, and the prediction effect may not be good on datasets with a small number of features. He Xiaowei et al. [22] developed a prediction model combining GRU and LSTM. Li Haoyang et al. [23] proposed an Informer-DCR model based on the improved Informer, where dilated causal convolution is introduced to enhance the prediction accuracy and ensure causality, thus improving the prediction accuracy.

The studies of [18–23] mainly focused on the innovative combinations of modelling architectures, ignoring the negative impact of extreme values in the data on model training. In dynamically changing environments, such as cloud resource management, extreme values are rare but often have a significant impact on system performance. Due to the atypical nature and unpredictable occurrences of these extreme values, predictive models are not only required to be able to capture and predict regular data trends, but also to acutely recognize and accurately predict extreme situations. While the simple combination of existing models has been effective in improving the overall prediction performance, such an approach does not pay sufficient attention to, or effectively handle, the occurrence of extreme values.

Therefore, this study aims to explore and implement a new prediction model that not only achieves the best overall prediction performance, but is also specifically optimized to improve the identification and prediction of extreme values in the data, with a view to provide more accurate and reliable CPU utilization prediction in a dynamic and complex container cluster environment.

## 3. Load Prediction Model

### 3.1. Description of the Problem

Container load prediction is a time-series data analysis task with the goal of estimating the CPU resource usage of the inner container in a given period. The data contain most of the normal values that contribute significantly to the overall predictability, as well as a few extreme values that must be accurately predicted. In addition, considering that container load is affected by a variety of factors, such as memory usage, disk occupancy, etc., we include these factors in our definition of container load data.

The problem can be described as follows:

For container load prediction, we set up a k-dimensional time series $\left( X_t = \left[ x_t^1, x_t^2, \ldots, x_t^k \right] \right)$, where the components represent different system metrics, such as CPU utilization $\left( x_t^1 \right)$, memory usage $\left( x_t^2 \right)$, and disk usage $\left( x_t^k \right)$. The prediction model $f$ utilizes historical observations $(X_{t-n+1}, \ldots, X_t)$ to estimate the CPU utilization $\left( \widehat{x_{t+1}^1} \right)$ at the next point in time $(t + 1)$.

$$\widehat{x_{t+1}^1} = f(X_{t-n+1}, \ldots, X_t) \tag{1}$$

where $f$ maps the historical data to predict future CPU utilization and represents the core component in multivariate time series forecasting.

The choice to forecast data for only one point in time in the future, rather than multiple points in time, was primarily made to avoid the problem of error accumulation in long-term forecasts. Long-term forecasting can lead to an increase in error accumulation over time, affecting forecast accuracy. Therefore, focusing on the forecast at the next point in time minimizes errors and ensures the reliability of the forecast results.

### 3.2. Load Prediction Model Based on Isolated Forest

In order to accurately predict extreme values in container CPU utilization, the ExtremoNet model is presented in this paper. ExtremoNet consists of the following three separate models: a normal value model, used to train the normal values in the predicted data; an extreme value model, used to train the extreme values in the predicted data; and the classifier model, trained to detect whether a value is categorized as a normal value or an extreme value.

This model combines the isolated forest algorithm and the classification sub-model to detect and integrate the information on abnormal extreme values to improve the accuracy of predicting extreme patterns. By introducing a classification sub-model, ExtremoNet can more accurately distinguish between normal and extreme events, ensuring that it remains efficient and accurate in a variable container load prediction environment. In addition, a temporal convolutional network (TCN) is used to sensitively capture short-term peak variations and minor fluctuations in a CPU to enhance the granularity and accuracy of the predictions.

Figure 1 illustrates the structure of the proposed ExtremoNet model. The model uses time-series data to estimate anomaly scores for each data point via an isolated forest algorithm, and then combines these scores with the raw data as features for model training.

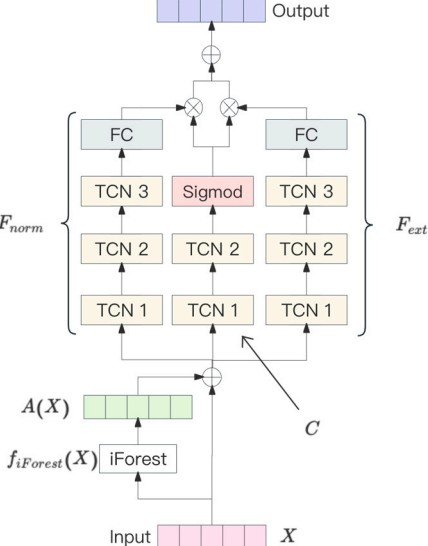

**Figure 1.** ExtremoNet model structure diagram.

The data flow is assigned to the following three paths in the model design: the extreme value prediction branch focuses on rare but important extreme load cases, the normal value prediction branch handles the constant load, and the auxiliary classification network is responsible for distinguishing between these two cases and improving the accuracy of the overall model in classifying the fluctuating nature of the data. The final prediction is accomplished by integrating the outputs of the two sub-prediction models and the decisions of the classification network. If the classification network judges the current data point as being extreme, the model focuses on the output; if it judges it to be normal, the output of the normal-value model is given a higher weight. The advantage of this approach is that it allows the model to dynamically adjust its focus on different types of loads, thus improving the overall accuracy of the prediction.

To describe the load prediction model in detail, its mathematical definition is first given. Let the time series data be $X_t$, where $t$ denotes the time step, and the goal of model $F$ is to predict the load at the next time step $\hat{X_{t+1}}$. Model $F$ contains three sub-models, $F_{ext}$, $F_{norm}$, and the auxiliary classification network $C$.

The mathematical representation of the model can be summarized as follows:

First, the function for calculating the abnormal score for the isolated forest algorithm is defined as follows:

$$A(X) = f_{iForest}(X) \tag{2}$$

The anomaly scores are combined as new data features to the original data as follows:

$$F(X) = X, A(X) \tag{3}$$

The classification network determines whether a data point is a normal or extreme fluctuation, giving probabilities $P_{norm}$ and $P_{ext}$.

$$P_{norm}(X), P_{ext}(X) = C(F(X)) \tag{4}$$

Then, based on the output of the classification network, the appropriate prediction model is selected as follows:

$$Y = P(X) = P_{norm}(X) \cdot F_{norm}(F(X)) + P_{ext}(X) \cdot F_{ext}(F(X)) \tag{5}$$

Ultimately, the output of the model Y is a weighted sum of the outputs of the two models.

In the above mathematical framework, model $F$ is able to adaptively predict both extreme and normal values. In predicting loads, the model not only captures daily fluctuation patterns, but also reacts to sudden extremes, thus enabling efficient adaptation to uncertain environments. This flexibility comes from the integrated design of the classification network $C$, $F_{ext}$, and $F_{norm}$, which allows the model to dynamically adjust the strength of its response to different load characteristics when making predictions. The fully connected layer downscales the data to integrate the information, and the Sigmoid activation function converts the classification result to a value between 0 and 1 for binary evaluation.

### 3.3. Evaluation Indicators

In this study, in order to evaluate the performance of the proposed ExtremoNet model in predicting the CPU load, we choose the following three commonly used evaluation metrics: the mean squared error (MSE), mean absolute error (MAE), and coefficient of determination ($R^2$). These metrics provide a comprehensive picture of the accuracy and stability of the model in predicting loads.

$$\text{MSE} = \frac{1}{n}\sum_{i=1}^{n}(y_i - \widehat{y_i})^2 \tag{6}$$

$$\text{MAE} = \frac{1}{n}\sum_{i=1}^{n}|y_i - \widehat{y_i}| \tag{7}$$

$$R^2 = 1 - \frac{\sum_{i=1}^{n}(y_i - \widehat{y_i})^2}{\sum_{i=1}^{n}(y_i - \overline{y})^2} \tag{8}$$

where $y_i$ is the true value, $\hat{y}_i$ is the predicted value, and n is the total number of samples.

By calculating these evaluation metrics in our experiments, we can fully evaluate the prediction performance of the proposed ExtremoNet model and compare it with other models. We hope to select the model with the best performance in predicting the CPU load by using these metrics.

## 4. Experimental Results and Analysis

### 4.1. Experimental Environment

In this study, the experimental environment was the Pytorch 1.7.1 framework, the operating system was the CentOS 7 system, Python version 3.8 was used, the system had 32 GB of memory and an NVIDIA Tesla T4 GPU, and the cudatoolkit version 11.3 was employed.

In this experiment, the batch_size, number of TCN channels, learning rate, and optimizer were set as described below. In order to ensure the optimal choice of hyperparameters for the algorithm, this study utilized the Optuna hyperparameter optimization framework to tune the model, where 80% of the data from each container was used as the training set and the remaining 20% as the test set. A wide range of hyperparameters, including the batch_size, number of TCN channels, and learning rate and optimizer, were searched for by Optuna. Using this method, the best combination of hyperparameters can be found automatically in a wide range of parameter spaces. The optimized parameters are as follows: the batch_size is 256, the number of TCN channels is [64,64,64], the learning rate is 0.0069, the MSE is used as the loss function, and Adam is chosen as the optimizer with an epoch of 100.

### 4.2. Experimental Data

The dataset for this experiment is from the AliCloud Platform trace dataset released in 2018.

The Cluster-trace-v2018 dataset [24], released by AliCloud Platform, provides trace data for about 4000 machines over 8 consecutive days. The dataset covers the partial workloads of these machines and the workloads of the entire cluster. All machines in the cluster run online tasks along with batch tasks. The dataset contains six types of tables, namely, machine_meta.csv, machine_usage.csv, container_meta.csv, container_usage.csv, batch_instance (batch_instance.csv), and batch_task.csv. In this paper, we mainly used the container utilization table and randomly extracted resource utilization data from 100 containers on day 4. The dataset field descriptions are shown in Table 1.

**Table 1.** AliCloud dataset field description table.

| Field | Type | Note |
|-------|------|------|
| cpu_util_percent | BIGINT | The current CPU utilization of the container |
| mem_util_percent | BIGINT | The current memory utilization of the container |
| cpi | DOUBLE | Number of cycles per instruction |
| mem_gps | DOUBLE | Memory bandwidth utilization |
| mpki | BIGINT | Number of out-of-page interrupts per thousand instructions |
| net_in | DOUBLE | Percentage of network inbound traffic utilization |
| net_out | DOUBLE | Percentage of network outgoing traffic utilization |
| disk_io_percent | DOUBLE | IO usage of the disk |

Figures 2 and 3 reveal the key features of the CPU utilization time series data from the AliCloud dataset. From Figure 2, it can be observed that there is significant volatility in the CPU utilization, with no shortage of spike–peak phenomena. This finding suggests that the system may have experienced a sudden increase in resource demand at certain moments.

In addition, the irregularity of the utilization fluctuation implies the dynamically changing nature of the load. In the histogram depicting the CPU utilization distribution, shown in Figure 3, the data clearly show a right-skewed distribution, where the high frequency of low-utilization intervals reflects that the system is under a lower load most of the time. However, the extension of the long tail in the graph indicates that high utilization extreme values, though rare, do exist and can have a significant impact on the operational stability of the system and resource scheduling strategies. Correctly predicting these extreme values is critical to the efficiency and reliability of cloud resource management systems. Therefore, the data characteristics presented in Figures 2 and 3 provide important guidance for predictive model design.

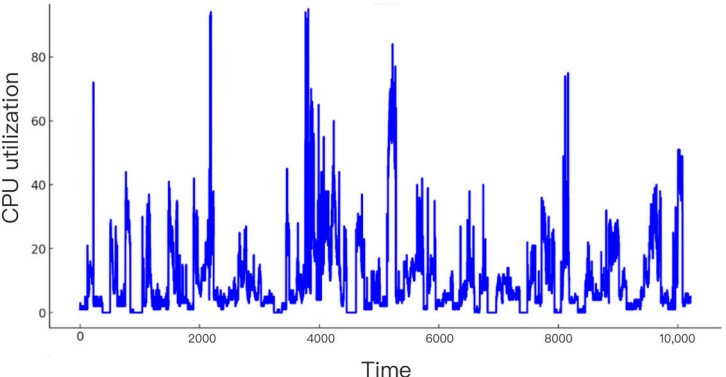

**Figure 2.** AliCloud dataset CPU utilization.

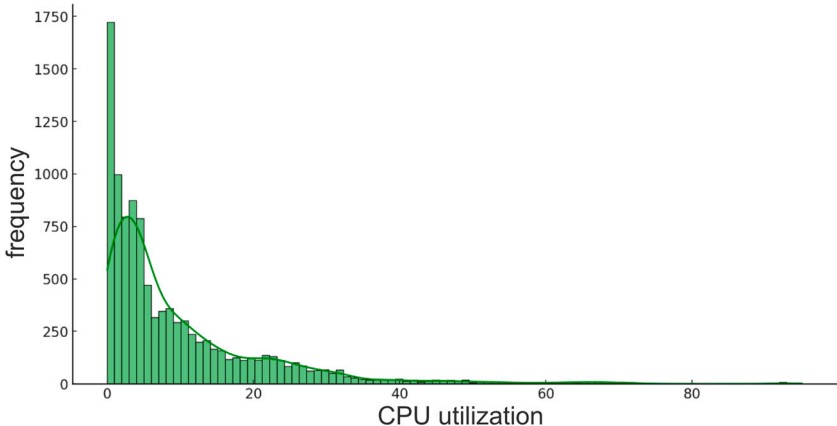

**Figure 3.** Frequency distribution of CPU utilization in AliCloud datasets.

### 4.3. Data Processing

Prior to further data processing, this study conducted a thorough quality assessment of the original AliCloud dataset to identify and correct potential data issues. Upon close inspection, the dataset was found to have missing data. To address the missing data and other problems in the dataset, we performed the following steps:

(1) Missing value treatment: To ensure data integrity, the Lagrange interpolation [25] was used in this study to supplement the missing values. The Lagrange interpolation utilizes all known data points to construct the interpolation polynomial, ensuring that the polynomial passes through every known data point. This global property makes it possible to capture the overall trend of the data series, not just the localized changes.

(2) Normalization: For preprocessing, MinMaxScaler was used to perform the feature normalization to eliminate the influence of the magnitude between different features in the

data and to improve the model training effect. MinMaxScaler can scale the feature values to a specified range, such as between 0 and 1, and its formula is expressed as

$$X_{scaled} = \frac{X - X_{min}}{X_{max} - X_{min}} \tag{9}$$

where $X$ represents the input feature vector and $X_{min}$ and $X_{max}$ represent the minimum and maximum values in the feature vector, respectively. After MinMaxScaler processing, the data features will be in the range from 0 to 1, which is conducive to model training and optimization.

(3) Extreme value determination: In order to determine whether a data point is an extreme value when training the model, the isolated forest algorithm is used to determine data points belonging to extreme values.

Figure 4 shows the outlier detection results in the CPU utilization data using the isolated forest algorithm. Each point in the figure represents an observation in the dataset, where the horizontal coordinate represents the index of the data point, and the vertical coordinate represents the corresponding CPU utilization rate.

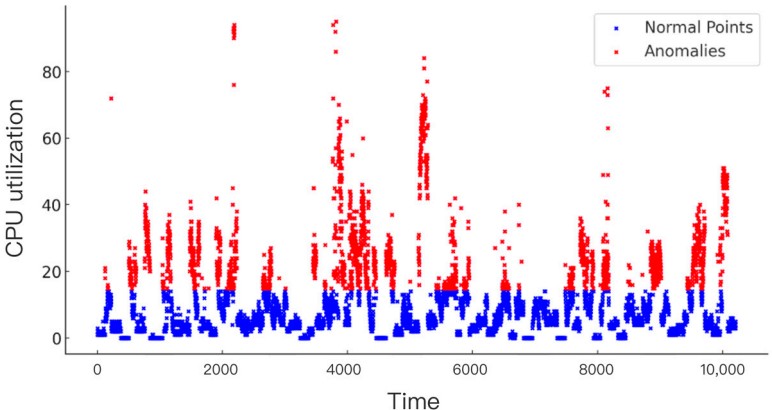

**Figure 4.** Detection of CPU utilization outliers based on the isolated forest algorithm.

The blue points in Figure 4 represent data points recognized as normal by the isolated forest algorithm. The CPU usage of these points is similar to the majority of the data points, indicating that their behavior is consistent with the overall distribution pattern of the dataset. The red points represent data points that are recognized as abnormal by the isolated forest algorithm. They are significantly different from the other data points in terms of CPU utilization and are therefore considered extreme values by the algorithm. These extreme values may indicate unusual or atypical behavior in the data.

(4) Feature Selection: In order to effectively select the features to be used for predicting the target variable cpu_usage, this paper uses the Pearson correlation coefficient. This method identifies the features that have the most influence on the prediction results by calculating the degree of linear correlation between each feature and the target variable.

Figure 5 shows the correlation coefficients of each feature with CPU_USAGE, and the importance of each feature is assessed based on the magnitude of the absolute value of these coefficients. Features with high correlation coefficients indicate a strong linear association with the target variable. Using this method, the most useful features for predicting CPU_USAGE are filtered from multiple features. From the above results, we can see that mem_gps has a low correlation coefficient, so we remove the mem_gps features from the data and all other features can be used for CPU utilization prediction.

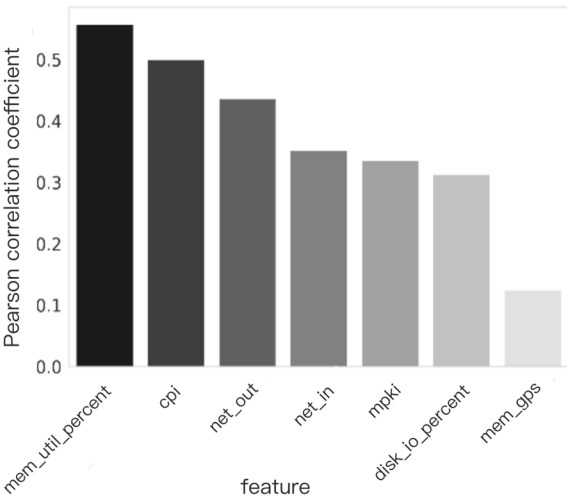

**Figure 5.** Correlation coefficients between different features in the dataset and the target feature.

*4.4. Comparison Experiment*

In the experiments, the time step is set to {2, 6, 12, 24} time units, i.e., cloud load data from the past {2, 6, 12, 24} time units are used as input features. Based on these input features, the model will predict the cloud loading scenarios, while three evaluation metrics, namely, the MSE, MAE, as well as R-squared values, are used to evaluate the model's performance.

Table 2 and Figure 6 clearly show the performance of each model on the AliCloud test dataset. ExtremoNet performs best compared to a single model. The experimental results show that the ExtremoNet model achieves the best performance on the AliCloud dataset. The MSE of the model is 7.79 and the $R^2$ is 96.51%.

**Table 2.** Experimental results for Alibaba Cloud dataset.

| Model | Step | 2 | 6 | 12 | 24 |
|---|---|---|---|---|---|
| ExtremoNet | MSE | 9.23 | 8.62 | 7.79 | 8.53 |
| | MAE | 1.53 | 1.66 | 1.44 | 1.48 |
| | $R^2$(%) | 94.10 | 94.75 | 96.51 | 95.89 |
| ARIMA | MSE | 22.21 | 19.17 | 16.07 | 15.13 |
| | MAE | 2.88 | 2.40 | 2.31 | 2.19 |
| | $R^2$(%) | 84.93 | 85.01 | 87.82 | 88.01 |
| TCN | MSE | 20.15 | 13.20 | 12.94 | 12.46 |
| | MAE | 2.70 | 1.91 | 1.88 | 1.85 |
| | $R^2$(%) | 85.33 | 90.43 | 90.64 | 90.83 |
| LSTM | MSE | 19.38 | 14.15 | 13.93 | 13.17 |
| | MAE | 2.49 | 2.07 | 2.21 | 2.17 |
| | $R^2$(%) | 85.89 | 89.74 | 89.92 | 90.21 |
| GRU | MSE | 16.32 | 16.44 | 13.79 | 12.57 |
| | MAE | 2.37 | 2.2 | 1.94 | 1.84 |
| | $R^2$(%) | 88.12 | 88.11 | 90.00 | 90.46 |

In this paper, the proposed model is compared with the GRU-LSTM model proposed in [22], the pCNN-LSTM model proposed in [13], and the CNN-BiGRU-Attention model proposed in [14]. The prediction results are shown in Figure 7, and the results of the evaluation metrics are shown in Table 3.

On the AliCloud dataset, the ExtremoNet model significantly outperforms the other models. For a 12-step length, the MSE of ExtremoNet is 7.79, while the MSEs of GRU-LSTM, pCNN-LSTM, and CNN-BiGRU-Attention are 14.82, 11.85, and 15.54, respectively. Compared to GRU-LSTM, pCNN-LSTM, and CNN-BiGRU-Attention, the MSE of ExtremoNet is reduced by 47.44%, 34.26%, and 49.87%. Similarly, ExtremoNet performs better in the

MAE metric, which is reduced by 23.40%, 24.21%, and 21.74% compared to GRU-LSTM, pCNN-LSTM, and CNN-BiGRU-Attention, respectively.

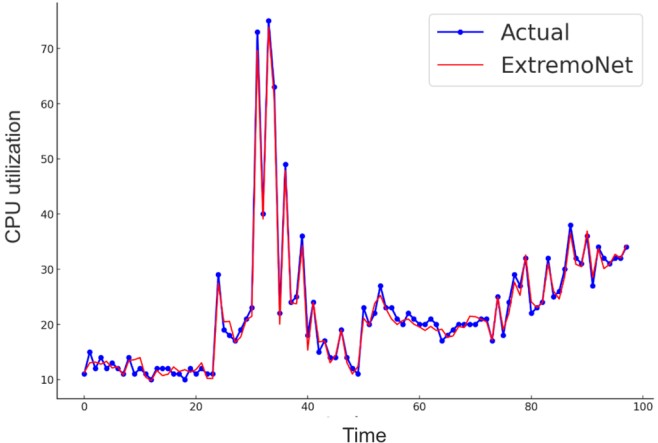

**Figure 6.** Predicted results on the Alibaba Cloud dataset.

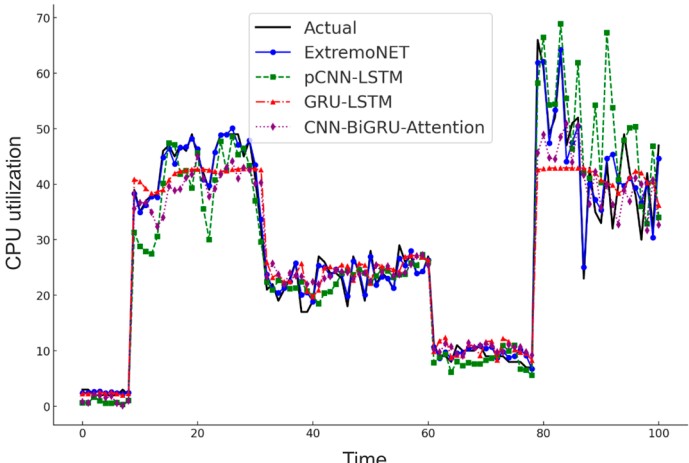

**Figure 7.** Plot comparing model prediction results.

**Table 3.** Comparison of the experimental results of the AliCloud dataset with recent methods.

| Model | Step | 2 | 6 | 12 | 24 |
|---|---|---|---|---|---|
| ExtremoNet | MSE | 9.23 | 8.62 | 7.79 | 8.53 |
| | MAE | 1.53 | 1.66 | 1.44 | 1.48 |
| | $R^2$(%) | 94.10 | 94.75 | 96.51 | 95.89 |
| GRU-LSTM | MSE | 17.24 | 13.42 | 14.82 | 13.21 |
| | MAE | 2.13 | 2.01 | 1.88 | 1.71 |
| | $R^2$(%) | 87.45 | 90.05 | 89.28 | 90.32 |
| pCNN-LSTM | MSE | 10.54 | 12.84 | 11.85 | 11.54 |
| | MAE | 1.86 | 1.94 | 1.90 | 1.86 |
| | $R^2$(%) | 92.35 | 90.69 | 91.42 | 91.69 |
| CNN-BiGRU-Attention | MSE | 15.60 | 14.89 | 15.54 | 14.92 |
| | MAE | 2.09 | 1.82 | 1.84 | 1.82 |
| | $R^2$(%) | 87.89 | 89.24 | 88.64 | 88.52 |

## 5. Conclusions

Given the limited ability of existing CPU resource prediction methods to deal with extreme CPU utilization values in cloud platforms, this study proposes an ExtremoNet model that incorporates the isolated forest algorithm and classification sub-models. The

model utilizes the isolated forest algorithm to identify and quantify extreme values and integrates the classification sub-model to further improve the model's ability to distinguish between normal and extreme fluctuations. Experiments conducted on the AliCloud dataset confirmed the effectiveness of our model, and it was compared with existing methods. The results show that ExtremoNet has significant advantages in its ability to recognize and predict extreme fluctuations, providing reliable support for container resource scheduling and decision making. It successfully bridges the gap of a single model in dealing with anomalous data, demonstrating efficiency and reliability in real-world applications.

Despite ExtremoNet's excellent prediction accuracy, there are still some challenges to be overcome. Due to the highly integrated structure of the model, its computational requirements are high, which may affect the responsiveness of the model in real-time prediction scenarios. In future work, we plan to explore ways to classify prediction intervals to further refine the prediction accuracy, and to consider more advanced techniques to reduce the computational burden of the model to ensure that it maintains its efficiency and accuracy in rapidly changing cloud computing environments.

**Author Contributions:** Conceptualization, C.W.; methodology, Z.W.; validation, Z.W.; investigation, Z.W.; data curation, Z.W.; writing—original draft preparation, Z.W. All authors have read and agreed to the published version of the manuscript.

**Funding:** This work was supported by the National Natural Science Foundation of China (No.62072363) and the Natural Science Foundation of Shaanxi Province (No. 2019JM-167).

**Institutional Review Board Statement:** Not applicable.

**Informed Consent Statement:** Not applicable.

**Data Availability Statement:** The data used in this study are available at ([https://github.com/alibaba/clusterdata/blob/master/cluster-trace-v2018](https://github.com/alibaba/clusterdata/blob/master/cluster-trace-v2018), accessed on 27 March 2024).

**Conflicts of Interest:** The authors declare no conflicts of interest.

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
