# Peer review of "Isolated Forest-Based Prediction of Container Resource Load Extremes"

_applsci, doi:10.3390/app14072911_

Round 1

Reviewer 1 Report

Comments and Suggestions for Authors

The study is about predicting CPU load in cloud computing and for it, an ML model based on iForest, named as ExtremoNet, is proposed. Although the proposal is useful, a lot of improvement in writing is needed. These are as follows:

Abstract:

- 'extreme values' must be defined, as one does not understand what type of extreme values the author is referring to. Instead, use 'extreme CPU load values'.

- remove 'data to enhance ... cases.'

- remove ', and enhance the ability ... pattern'. It is same as previous statement.

- 'enhancement in MAE', should be replaced by ' reduction in MAE'. Moreover rephrase the statement about MSE and MAE.

- Overall the abstract is not well-written and it must be improved.

1. Introduction:

- Mention, the benefit of predicting 'extreme values'.

- remove ' and flexibility' in line 55

- Organization of the paper should be added to the end of the Introduction.

2. Related Work

- what are 'other types of loads' (line 95)

- Rather than using the term 'Literature' repeatedly, used the initials of the author(s) of the articles.

3. 1:

- In equation (1) superscript with X, while in equation (2) subscript with X is used.

- remove lines 155 and 156, it is a repetition.

3.2

- In equation (1), and (2) and in text, there is no need to use subscript 't', as the first subscript with x is already denoting timestamp.

- In line 158, remove 't moment'.

- Different components of Figure 1 should be explained, such as FC.  

- The components of the model, mentioned in lines 190-194, are not shown in Figure 1.

- ',' is not an operator and is never used in right hand side of an equation, as is used in equation 4.

- rather that mentioning two separate equations in 6 and 7, combine them by placing 'Y' at the LHS of the equation 6. Also remove 202-203 lines

3.3

- remove the first statement in 211 line. It is a repetition

- Itemize the steps describing the process of iForest. State the goals of each of these three steps.

- There are some writing mistakes in line 220. Correct them.

- In line 214 'n' denotes the number of data points in the dataset while in 223 line, 'n' denotes sample number. This is creating confusion in terminology.

- remove 'Determination  (R2). in line 230.

4. Experimental results and analysis

4.1. heading should be capitalized

- remove the first statement in paragraph 2, it has both typos and is repeated later in the same paragraph.

4.2.

- move the mention of the github repository in line 255, either to footnote or to references.

- there are typos in line 261 and 263.

- At some places 'Fig.' while in others 'Figure' is used. Be consistent.

4.3 heading should be capitalized

- Explain why  'Lagrangian interpolation' is preferred.

- Remove line 303. 

- Add '.' at the end of every figure caption.

- Correct the alignment of x-axis labels in Figure 5.

- Use the correct caption in Figure 5. It should be similar to the 'correlation of features with the target feature'.

- Replace 'other features' with 'rest of the features' in line 330.

4.4. heading should be capitalized

- Both table 2 and table 3 depict similar information. They must be merged into one table. 

5. Conclusion (omit 's' at the end).

- In line 369, replace 'in several aspects' with 'accuracy', as there is only an accuracy comparison in the paper.

- In line 372, replace 'this paper plans' to 'we plan'.

I suggest that the computation time of the proposed method should also be compared with the baseline methods. 

Comments on the Quality of English Language

The paper is understandable, but there are a lot of typos, that need correction.

Reviewer 2 Report

Comments and Suggestions for Authors

This paper introduce a CPU utilizatiopn prediction model for container scheduler. The prediction use isolated forest algorithm to determine if data is an anomaly. Extreme data (anomaly) is then predicted using extreme value prediction branch. Non-extreme value are handled by normal prediction branch. 

While the idea is interesting, the publication require more works before publish.

Main criticisms:

1) The experiments is done on a single dataset, where 80% of the data is used for training and 20% is used for evaluating.

It can be argued that the same data set may contain certain characteristics which makes the 20% test set track closely similar to the 80% training set.

2) This work require at least one additional polishing and proof reading.

Example of minor proof reading points:

 Figures and their captions are not centered on the page

 Page 4: line 153-154 and 155-156 are (partially) duplicated. Same for page 8, line 303 and 304-307.

 Page 5: Formula 7 doesn't reflect text written on line 202. It doesn't show "weighted sum of outputs"

 Page 6: line 220 seems to have unformatted latex code "s\left(x_i,n\right)".

 Page 6: line 226. "Evaluation indicators" -> v letter is not in a normal font

 Page 6: line 222, "𝑐(𝑛)is" -> need spacing before "is"

Other trivial observation which doesn't affect the decision:

 Experiment is done on CentOs 7 which was released in 2014. Currently is on the last half year of Maintenance update period. This does not affect the current accuracy experiments. But maybe relevant in the future if the algorithm performance/overhead is to be evaluated.

Round 2

Reviewer 1 Report

Comments and Suggestions for Authors

Although the authors has improved their writing still three things need correction.

1. Do not use 'was' for the current work in the abstract. The word 'is' is fine.

2. Instead of just using the word 'extreme', replace it with 'extreme values'.

3. The names of the ML models should be capitalized.

Most importantly in the revised version, there is a lot of strike through which makes reading hard. I recommend that authors should thoroughly go over the final version of the paper. Typos should be avoided.

Comments on the Quality of English Language

It has improved a lot, compared to the first version.

Author Response

Dear Reviewer,

Thank you for your constructive feedback. I have carefully addressed the issues you highlighted:

I have replaced 'was' with' is' in the abstract to reflect the current state of the work accurately.
The term 'extreme' has been amended to 'extreme values' throughout the text to enhance clarity.
All machine learning model names have been capitalized to adhere to the proper convention.

Additionally, I have proofread the manuscript to correct any typographical errors and formatting issues.

Thank you for your guidance, which has undoubtedly improved the manuscript's quality.

​Sincerely,
Zhenbang Wang

Reviewer 2 Report

Comments and Suggestions for Authors

Author Submitted change-highlighted publication where changes are highlighted in red.

This revision significantly improves the writing quality. Some sections become slightly shorter. However, we cannot evaluate the formatting since the version without change highlighting is not included for review.

Comments on the Quality of English Language

There are still minor correction to be made.
For example, line 291 page 7: Eesults -> Results

Author Response

Dear Reviewer,

Thank you for your valuable feedback on the manuscript.  I have carefully reviewed and addressed the formatting issues you mentioned.  I assure you that this version's formatting has been meticulously proofread and adjusted to meet the expected standards.

For your convenience and thorough evaluation, I suggest comparing this revised version with the initial submission to assess the improvements and changes in formatting clearly.

I appreciate your attention to detail and guidance, which have been instrumental in enhancing the quality of this manuscript.

Best regards,
Zhenbang Wang